# Deep Generative Video Compression

**Jun Han**[*]
Dartmouth College
junhan@cs.dartmouth.edu

**Salvator Lombardo**[*]
Disney Research LA
salvator.d.lombardo@disney.com

**Christopher Schroers**
DisneyResearch|Studios
christopher.schroers@disney.com

**Stephan Mandt**
University of California, Irvine
mandt@uci.edu

## Abstract

The usage of deep generative models for image compression has led to impressive performance gains over classical codecs while neural video compression is still in its infancy. Here, we propose an end-to-end, deep generative modeling approach to compress temporal sequences with a focus on video. Our approach builds upon variational autoencoder (VAE) models for sequential data and combines them with recent work on neural image compression. The approach jointly learns to transform the original sequence into a lower-dimensional representation as well as to discretize and entropy code this representation according to predictions of the sequential VAE. Rate-distortion evaluations on small videos from public data sets with varying complexity and diversity show that our model yields competitive results when trained on generic video content. Extreme compression performance is achieved when training the model on specialized content.

## 1 Introduction

The transmission of video content is responsible for up to 80% of the consumer internet traffic, and both the overall internet traffic as well as the share of video data is expected to increase even further in the future (Cisco, 2017). Improving compression efficiency is more crucial than ever. The most commonly used standard is H.264 (Wiegand et al., 2003); more recent codecs include H.265 (Sullivan et al., 2012) and VP9 (Mukherjee et al., 2015). All of these existing codecs follow the same block based hybrid structure (Musmann et al., 1985) which essentially emerged from engineering out and refining this concept over decades. From a high level perspective, they differ in a huge number of smaller design choices and have grown to become more and more complex systems.

While there is room for improving the block based hybrid approach even further (Fraunhofer, 2018), the question remains as to how much longer significant improvements can be obtained while following the same paradigm. In the context of image compression, deep learning approaches that are fundamentally different to existing codecs have already shown promising results (Ballé et al., 2018, 2016; Theis et al., 2017; Agustsson et al., 2017; Minnen et al., 2018). Motivated by these successes for images, we propose a first step towards innovating beyond block-based hybrid codecs by framing video compression in a deep generative modeling context. To this end, we propose an unsupervised deep learning approach to encoding video. The approach simultaneously learns the optimal transformation of the video to a lower-dimensional representation *and* a powerful predictive model that assigns probabilities to video segments, allowing us to efficiently entropy-code the discretized latent representation into a short code length.

---

[*] Shared first authorship.

H.265 (**21.1** dB @ **0.86** bpp)  VP9 (**26.0** dB @ **0.57** bpp)  Ours (**44.6** dB @ **0.06** bpp)

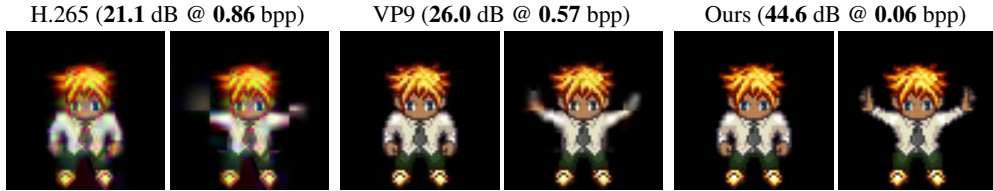

Figure 1:   Reconstructed video frames using the established codecs H.265 (left), VP9 (middle), and ours (right), with videos taken from the Sprites data set (Section 4). On specialized content as shown here, higher PSNR values in dB (corresponding to lower distortion) can be achieved at almost an order of magnitude smaller bits per pixel (bpp) rates. Compared to the classical codecs, fewer geometrical artifacts are apparent in our approach.

Our end-to-end neural video compression scheme is based on sequential variational autoencoders (Bayer & Osendorfer, 2014; Chung et al., 2015; Li & Mandt, 2018). The transformations to and from the latent representation (the encoder and decoder) are parametrized by deep neural networks and are learned by unsupervised training on videos. These latent states have to be discretized before they can be compressed into binary. Ballé et al. (2016) address this problem by using a box-shaped variational distribution with a fixed width, forcing the VAE to 'forget' all information stored on smaller length scales due to the insertion of noise during training. This paper follows the same paradigm for temporally-conditioned distributions. A sequence of quantized latent representations still contains redundant information as the latents are highly correlated. (Lossless) entropy encoding exploits this fact to further reduce the expected file size by expressing likely data in fewer bits and unlikely data in more bits. This requires knowledge of the probability distribution over the discretized data that is to be compressed, which our approach obtains from the sequential prior.

Among the many architectural choices that our approach enables, we empirically investigate a model that is well suited for the regime of extreme compression. This model uses a combination of both *local* latent variables, which are inferred from a single frame, and a *global* state, inferred from a multi-frame segment, to efficiently store a video sequence. The dynamics of the local latent variables are modeled stochastically by a deep generative model. After training, the context-dependent predictive model is used to entropy code the latent variables into binary with an arithmetic coder.

In this paper, we focus on low-resolution video ($64 \times 64$) as the first step towards deep generative video compression. Figure 1 shows a test example of the possible performance improvements using our approach if the model is trained on restricted content (video game characters). The plots show two frames of a video, compressed and reconstructed by our approach and by classical video codecs. One sees that fine granular details, such as the hands of the cartoon character, are lost in the classical approach due to artifacts from block motion estimation (low bitrate regime), whereas our deep learning approach successfully captures these details with less than 10% of the file length.

Our contributions are as follows:

**1) A general paradigm for generative compression of sequential data.**  We propose a general framework for compressing sequential data by employing a sequential variational autoencoder (VAE) in conjuction with discretization and entropy coding to build an end-to-end trainable codec.

**2) A new neural codec for video compression.** We employ the above paradigm towards building an end-to-end trainable codec. To the best of our knowledge, this is the first work to utilize a deep generative video model together with discretization and entropy coding to perform video compression.

**3) High compression ratios.** We perform experiments on three public data sets of varying complexity and diversity. Performance is evaluated in terms of rate-distortion curves. For the low-resolution videos considered in this paper, our method is competitive with traditional codecs after training and testing on a diverse set of videos. Extreme compression performance can be achieved on a restricted set of videos containing specialized content if the model is trained on similar videos.

**4) Efficient compression from a global state.** While a deep latent time series model takes temporal redundancies in the video into account, one optional variation of our model architecture tries to compress static information into a separate global variable (Li & Mandt, 2018) which acts similarly as a key frame in traditional methods. We show that this decomposition can be beneficial.

Our paper is organized as follows. In Section 2, we summarize related works before describing our method in Section 3. Section 4 discusses our experimental results. We give our conclusions in Section 5.

## 2  Related Work

The approaches related to our method fall into three categories: deep generative video models, neural image compression, and neural video compression.

**Deep generative video models.**  Several works have applied the variational autoencoder (VAE) (Kingma & Welling, 2014; Rezende et al., 2014) to stochastically model sequences (Bayer & Osendorfer, 2014; Chung et al., 2015). Babaeizadeh et al. (2018); Xu et al. (2020) use a VAE for stochastic video generation. He et al. (2018) and Denton & Fergus (2018) apply a long short term memory (LSTM) in conjunction with a sequential VAE to model the evolution of the latent space across many video frames. Li & Mandt (2018) separate latent variables of a sequential VAE into local and global variables in order to learn a disentangled representation for video generation. Vondrick et al. (2016) generate realistic videos by using a generative adversarial network (Goodfellow et al., 2014) to learn to separate foreground and background, and Lee et al. (2018) combine variational and adversarial methods to generate realistic videos. This paper also employs a deep generative model to model the sequential probability distribution of frames from a video source. In contrast to other work, our method learns a continuous latent representation that can be discretized with minimal information loss, required for further compression into binary. Furthermore, our objective is to convert the original video into a short binary description rather than to generate new videos.

**Neural image compression.**   There has been significant work on applying deep learning to image compression. In Toderici et al. (2016, 2017); Johnston et al. (2018), an LSTM based codec is used to model spatial correlations of pixel values and can achieve different bit-rates without having to retrain the model. Ballé et al. (2016) perform image compression with a VAE and demonstrate how to approximately discretize the VAE latent space by introducing noise during training. This work is refined by (Ballé et al., 2018) which improves the prior model (used for entropy coding) beyond the mean-field approximation by transmitting side information in the form of a hierarchical model. Minnen et al. (2018) consider an autoregressive model to achieve a similar effect. Santurkar et al. (2018) studies the performance of generative compression on images and suggests it may be more resilient to bit error rates. These image codecs encode each image independently and therefore their probabilistic models are stationary with respect to time. In contrast, our method performs compression according to a non-stationary, time-dependent probability model which typically has lower entropy per pixel.

**Neural video compression.**   The use of deep neural networks for video compression is relatively new. Wu et al. (2018) perform video compression through image interpolation between reference frames using a predictive model based on a deep neural network. Chen et al. (2017) and Chen et al. (2019) use a deep neural architecture to predict the most likely frame with a modified form of block motion prediction and store residuals in a lossy representation. Since these works are based on motion estimation and residuals, they are somewhat similar in function and performance to existing codecs. Lu et al. (2019) and Djelouah et al. (2019) also follow a pipeline based on motion estimation and residual computation as in existing codecs. In contrast, our method is not based on motion estimation, and the full inferred probability distribution over the space of plausible subsequent frames is used for entropy coding the frame sequence (rather than residuals). In a concurrent publication, Habibian et al. (2019) perform video compression by utilizing a 3D variational autoencoder. In this case, the 3D encoder removes temporal redundancy by decorrelating latents, wheras our method uses entropy coding (with time-dependent probabilities) to remove temporal redundancy.

## 3  Deep Generative Video Compression

Our end-to-end approach simultaneously learns to transform a video into a lower-dimensional latent representation *and* to remove the remaining redundancy in the latents through model-based entropy coding. Section 3.1 gives an overview of the deep generative video coding approach as a whole

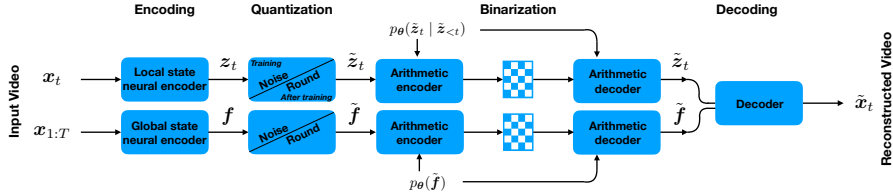

Figure 2: High-level operational diagram of our compression codec (see Section 3). A video segment is encoded into per-frame latent variables $z_t$ and (optionally) also into a per-segment global state $f$ using a VAE architecture. Both latent variables are then quantized and arithmetically encoded into binary according to the respective prior models. To recover an approximation to the original video, the latent variables are arithmetically decoded from the binary and passed through the neural decoder.

before Sections 3.2 and 3.3 detail on the model-based entropy coding and the lower-dimensional representation, respectively.

## 3.1 Overview

Lossy video compression is a constrained optimization problem that can be approached from two different angles: 1) either as finding the shortest description of a video without exceeding a certain level of information loss or 2) as finding the minimal level of information loss without exceeding a certain description length. Both optimization problems are equivalent with either a focus on description length (rate) or information loss (distortion) constraints. The distortion is a measure of how much error encoding and subsequent decoding incurs while the rate quantifies the amount of bits the encoded representation occupies. When denoting distortion by $\mathcal{D}$, rate by $\mathcal{R}$, and the maximal rate constraint by $\mathcal{R}_c$, the compression problem can be expressed as

$$\min \mathcal{D} \quad \text{subject to} \quad \mathcal{R} \leq \mathcal{R}_c.$$

Such a constrained formulation is often cumbersome but can be solved in a Lagrange multiplier formulation, where the rate and distortion terms are weighted against each other by a Lagrange multiplier $\beta$:

$$\min \mathcal{D} + \beta \mathcal{R}. \tag{1}$$

In existing video codecs, encoders and decoders have been meticulously engineered to improve coding efficiency. Instead of engineering encoding and decoding functions, in our end-to-end machine learning approach we aim to *learn* these mappings by parametrizing them by deep neural networks and then optimizing Eq. 1 accordingly.

There is a well-known equivalence (Ballé et al., 2018; Alemi et al., 2018) between the evidence lower bound in amortized variational inference (Gershman & Goodman, 2014; Zhang et al., 2018), and the Lagrangian formulation of lossy coding of Eq. 1. Variational inference involves a probabilistic model $p(\boldsymbol{x}, \boldsymbol{z}) = p(\boldsymbol{x}|\boldsymbol{z})p(\boldsymbol{z})$ over data $\boldsymbol{x}$ and latent variables $\boldsymbol{z}$. The goal is to lower-bound the marginal likelihood $p(\boldsymbol{x})$ using a variational distribution $q(\boldsymbol{z}|\boldsymbol{x})$. When the variational distribution $q$ has a fixed entropy (e.g., by fixing its variance), this bound is, up to a constant,

$$\mathbb{E}_q[\log p(\boldsymbol{x}|\boldsymbol{z})] - \beta \, H[q(\boldsymbol{z}|\boldsymbol{x}), p(\boldsymbol{z})], \tag{2}$$

where $H$ is the cross entropy between the approximate posterior and the prior. When allowing for arbitrary $\beta$, Ballé et al. (2016) showed in the context of image compression with variational autoencoders that the negative of Eq. 2 becomes a manifestation of Eq. 1. While the first term measures the expected reconstruction error of the encoded images, the cross entropy term becomes the expected code length as the (learned) prior $p(\boldsymbol{z})$ is used to inform a lossless entropy coder about the probabilities of the discretized encoded images. In this paper we generalize this approach to videos by employing probabilistic deep sequential latent state models.

Fig. 2 summarizes our overall design. Given a sequence of frames $\boldsymbol{x}_{1:T} = (\boldsymbol{x}_1, \ldots, \boldsymbol{x}_T)$, we transform them into a sequence of latent states $\boldsymbol{z}_{1:T}$ and optionally also a global state $\boldsymbol{f}$. Although this transformation into a latent representation is lossy, the video is not yet optimally compressed as there are still correlations in the latent space variables. To remove this redundancy, the latent

space must be entropy coded into binary. This is the distinguishing element between variational autoencoders and full compression algorithms. The bit stream can then be sent to a receiver where it is decoded into video frames. Our end-to-end machine learning approach simultaneously learns the predictive model required for entropy coding *and* the optimal lossy transformation into the latent space. Both components are described in detail in the next sections, respectively.

## 3.2 Entropy Coding via a Deep Sequential Model

Predictive modeling is crucial at the entropy coding stage. A better model which more accurately captures the true certainty about the next symbol has a smaller cross entropy with the data distribution and thus produces a bit rate that is closer to the theoretical lower bound for long sequences (Shannon, 2001). For videos, temporal modeling is most important, making a learned temporal model an integral part of our model design. We now discuss a preliminary version of our model which does not yet include the global state, saving the specific details and encoder of our proposed model for Section 3.3.

**General model design.** When it comes to designing a generative model, the challenge over image compression is that videos exhibit strong temporal correlations in addition to the spatial correlations present in images. Treating a video segment as an independent data point in the latent representation (as would a 3D autoencoder) leads to data sparseness and poor generalization performance. Therefore, we propose to learn a temporally-conditioned prior distribution parametrized by a deep generative model to efficiently code the latent variables associated with each frame. Let $\boldsymbol{x}_{1:T} = (\boldsymbol{x}_1, \cdots, \boldsymbol{x}_T)$ be the video sequence and $\boldsymbol{z}_{1:T}$ be the associated latent variables. A generic generative model of this type takes the form:

$$p_{\boldsymbol{\theta}}(\boldsymbol{x}_{1:T}, \boldsymbol{z}_{1:T}) = \prod_{t=1}^{T} p_{\boldsymbol{\theta}}(\boldsymbol{z}_t | \boldsymbol{z}_{<t}) p_{\boldsymbol{\theta}}(\boldsymbol{x}_t \mid \boldsymbol{z}_t). \tag{3}$$

Above, $\boldsymbol{\theta}$ is shorthand for parameters of the model. By conditioning on previous frame latents in the sequence, the prior model can be more certain about the next $\boldsymbol{z}_t$, thus achieving a smaller entropy and code length (after entropy coding).

**Arithmetic coding.** Entropy coding schemes require a discrete vocabulary, which is obtained in our case by rounding the latent states to the nearest integer after training. Care must be taken such that the quantization at inference time is approximated in a differentiable way during training. In practice, this is handled by introducing noise in the inference process. Besides dealing with quantization, we also need an accurate estimate of the probability density over the latent atoms for efficient coding. Knowledge of the sequential probability distribution of latents allows the entropy coder to decorrelate the bitstream so that the maximal amount of information per bit is stored (MacKay, 2003). We obtain this probability estimation from the learned prior.

We employ an arithmetic coder (Rissanen & Langdon, 1979; Langdon, 1984) to losslessly code the rounded latent variables into binary. In contrast to other forms of entropy encoding, such as Huffman coding, arithmetic coding encodes the entire sequence of discretized latent states $\boldsymbol{z}_{1:T}$ into a single number. During encoding, the approach uses the conditional probabilities $p(\boldsymbol{z}_t | \boldsymbol{z}_{<t})$ to iteratively refine the real number interval $[0, 1)$ into a progressively smaller interval. After the sequence has been processed and a final (very small) interval is obtained, a binarized floating point number from the final interval is stored to encode the entire sequence of latents. Decoding the decimal can similarly be performed iteratively by undoing the sequence of interval refinements to recover the original latent sequence. The fact that decoding happens in the same temporal order as encoding guarantees access to all conditional probabilities $p(\boldsymbol{z}_t | \boldsymbol{z}_{<t})$. Since $\boldsymbol{z}_t$ was quantized, all probabilities for encoding and decoding exactly match. In practice, besides iterating over time stamps $t$, we also iterate over the dimensions of $\boldsymbol{z}_t$ during arithmetic coding.

## 3.3 Proposed Generative and Inference Model

In this section, we describe the modeling aspects of our approach in more detail. We refine the generative model to also include a global state which can be omitted to capture the base case outlined before. Besides the local state, the global state may help the model capture long-term information.

**Decoder.** The decoder is a probabilistic neural network that models the data as a function of their underlying latent codes. We use a stochastic recurrent variational autoencoder that transforms a sequence of local latent variables $z_{1:T}$ and a global state $f$ into the frame sequence $x_{1:T}$, expressed by the following joint distribution:

$$p_{\theta}(x_{1:T}, z_{1:T}, f) = p_{\theta}(f) p_{\theta}(z_{1:T}) \prod_{t=1}^{T} p_{\theta}(x_t \mid z_t, f). \tag{4}$$

We discuss the prior distributions $p_{\theta}(f)$ and $p_{\theta}(z_{1:T})$ separately below. Each reconstructed frame $\tilde{x}_t$, sampled from the frame likelihood $p_{\theta}(x_t|f, z_t)$, depends on the corresponding latent variables $z_t$ and (optionally) global variables $f$. We use a Laplace distribution for the frame likelihood, $p_{\theta}(x_t \mid z_t, f) = \text{Laplace}(\mu_{\theta}(z_t, f), \lambda^{-1}\mathbf{1})$, whose logarithm results in an $\ell_1$ loss which we observe produces sharper images than the $\ell_2$ loss (Isola et al., 2017; Zhao et al., 2016).

The decoder mean, $\mu_{\theta}(\cdot)$, is a function parametrized by neural networks. Crucially, the decoder is conditioned both on global code $f$ and time-local code $z_t$. In detail, $(f, z_t)$ are combined by a multilayer perceptron (MLP) which is then followed by upsampling transpose convolutional layers to form the mean. More details on the architecture can be found in the supplementary material. After training, the reconstructed frame in image space is obtained from the mean, $\tilde{x}_t = \mu_{\theta}(z_t, f)$.

**Encoder.** As the inverse of the decoder, the optimal encoder would be the Bayesian posterior $p(z_{1:T}, f \mid x_{1:T})$ of the generative model above, which is analytically intractable. Therefore, we employ amortized variational inference (Blei et al., 2017; Zhang et al., 2018; Marino et al., 2018) to predict a distribution over latent codes given the input video,

$$q_{\phi}(z_{1:T}, f \mid x_{1:T}) = q_{\phi}(f \mid x_{1:T}) \prod_{t=1}^{T} q_{\phi}(z_t \mid x_t). \tag{5}$$

The global variables $f$ are inferred from all video frames in a sequence and may thus contain global information, while $z_t$ is only inferred from a single frame $x_t$.

As explained above in Section 3.2, modifications to standard variational inference are required for further lossless compression into binary. Instead of sampling from Gaussian distributions with learned variances, here we employ fixed-width uniform distributions centered at their means: $\tilde{f} \sim q_{\phi}(f \mid x_{1:T}) = \mathcal{U}(\hat{f} - \frac{1}{2}, \hat{f} + \frac{1}{2})$, $\quad \tilde{z}_t \sim q_{\phi}(z_t \mid x_t) = \mathcal{U}(\hat{z}_t - \frac{1}{2}, \hat{z}_t + \frac{1}{2})$.

The means are predicted by additional encoder neural networks $\hat{f} = \mu_{\phi}(x_{1:T})$, $\hat{z}_t = \mu_{\phi}(x_t)$ with parameters $\phi$. This choice of inference distribution leads exactly to injection of noise with width one centered at the maximally-likely values for the latent variables, described in Section 3.2. The mean for the global state is parametrized by convolutions over $x_{1:T}$, followed by a bi-directional LSTM which is then processed by a MLP. The encoder mean for the local state is simpler, consisting of convolutions over each frame followed by a MLP. More details on the decoder architecture is provided in the supplementary material.

**Prior Models.** The models parametrizing the learned prior distributions are ultimately used as the probability models for entropy coding. The global prior $p_{\theta}(f)$ is assumed to be stationary, while $p_{\theta}(z_{1:T})$ consists of a time series model. Each dimension of the latent space has its own density model:

$$p_{\theta}(f) = \prod_{i}^{\dim(f)} p_{\theta}(f^i) * \mathcal{U}(-\frac{1}{2}, \frac{1}{2}); \qquad p_{\theta}(z_{1:T}) = \prod_{t}^{T} \prod_{i}^{\dim(z)} p_{\theta}(z_t^i \mid z_{<t}) * \mathcal{U}(-\frac{1}{2}, \frac{1}{2}). \tag{6}$$

Above, indices refer to the dimension index of the latent variable. The convolution with uniform noise is to allow the priors to better match the true marginal distribution when working with the box-shaped approximate posterior (see Ballé et al. (2018) Appendix 6.2). This convolution has an analytic form in terms of the cumulative probability density.

The stationary density $p_{\theta}(f^i)$ is adopted from (Ballé et al., 2018); it is a flexible non-parametric, fully-factorized model that leads to a good matching between prior and latent code distribution. The density is defined by its cumulative and is built out of compositions of nonlinear probability densities, similar to the construction of a normalizing flow (Rezende & Mohamed, 2015).

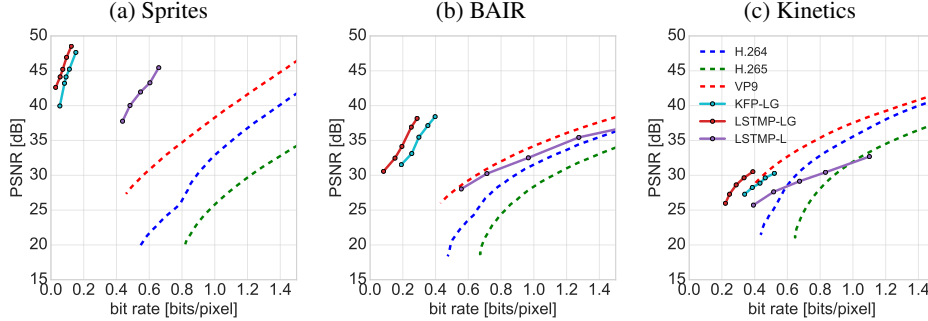

Figure 3: Rate-distortion curves on three datasets measured in PSNR (higher corresponds to lower distortion). Legend shared. Solid lines correspond to our models, with LSTMP-LG proposed.

Two dynamical models are considered to model the sequence $\boldsymbol{z}_{1:T}$. We propose a a recurrent LSTM prior architecture for $p_{\boldsymbol{\theta}}(z_t^i \mid \boldsymbol{z}_{<t})$ which conditions on all previous frames in a segment. The distribution $p_{\boldsymbol{\theta}}$ is taken to be normal with mean and variance predicted by the LSTM. We also considered a simpler model, which we compare against, with a single frame context, $p_{\boldsymbol{\theta}}(z_t^i \mid \boldsymbol{z}_{<t}) = p_{\boldsymbol{\theta}}(z_t^i \mid \boldsymbol{z}_{t-1})$, which is essentially a deep Kalman filter (Krishnan et al., 2015).

**Variational Objective.** The encoder (variational model) and decoder (generative model) are learned jointly by maximizing the $\beta$-VAE objective (Higgins et al., 2017; Mandt et al., 2016),

$$\mathcal{L}(\phi, \theta) = \mathbb{E}_{\tilde{\boldsymbol{f}}, \tilde{\boldsymbol{z}}_{1:T} \sim q_\phi}[\log p_{\boldsymbol{\theta}}(\boldsymbol{x}_{1:T}|\tilde{\boldsymbol{f}}, \tilde{\boldsymbol{z}}_{1:T})] + \beta \, \mathbb{E}_{\tilde{\boldsymbol{f}}, \tilde{\boldsymbol{z}}_{1:T} \sim q_\phi}[\log p_{\boldsymbol{\theta}}(\tilde{\boldsymbol{f}}, \tilde{\boldsymbol{z}}_{1:T})]. \qquad (7)$$

The first term corresponds to the distortion, while second term is the cross entropy between the approximate posterior and the prior. The latter has the interpretation of the expected code length when using the prior distribution $p(\boldsymbol{f}, \boldsymbol{z}_{1:T})$ to entropy code the latent variables. It is known (Hoffman & Johnson, 2016) that this term encourages the prior model to approximate the empirical distribution of codes, $\mathbb{E}_{\boldsymbol{x}_{1:T}}[q(\boldsymbol{f}, \boldsymbol{z}_{1:T}|\boldsymbol{x}_{1:T})]$. For our choice of generative model, the cross entropy separates into two independent terms $H[q_\phi(\boldsymbol{f}|\boldsymbol{x}_{1:T}), p_{\boldsymbol{\theta}}(\boldsymbol{f})]$ and $H[q_\phi(\boldsymbol{z}_{1:T}|\boldsymbol{x}_{1:T}), p_{\boldsymbol{\theta}}(\boldsymbol{z}_{1:T})]$. Note that for our choice of variational distribution, the entropy contribution of $q_\theta$ is constant and is therefore omitted.

# 4 Experiments

In this section, we present the experimental results of our work. We first describe the datasets, performance metrics, and baseline methods in Section 4.1. This is followed by a quantitative analysis in terms of rate-distortion curves in Section 4.2 which is followed by qualitative results in Section 4.3.

## 4.1 Datasets, Metrics, and Methods

In this work, we train separately on three video datasets of increasing complexity with frame size $64 \times 64$. **1) Sprites.** The simplest dataset consists of videos of Sprites characters from an open-source video game project, which is used in (Reed et al., 2015; Mathieu et al., 2016; Li & Mandt, 2018). The videos are generated from a script that samples the character action, skin color, clothing, and eyes from a collection of choices and have an inherently low-dimensional description (*i.e.* the script that generated it). **2) BAIR.** BAIR robot pushing dataset (Ebert et al., 2017) consists of a robot pushing objects on a table, which is also used in (Babaeizadeh et al., 2018; Denton & Fergus, 2018; Lee et al., 2018). The video is more realistic and less sparse, but the content is specialized since all scenes contain the same background and robot, and the depicted action is simple since the motion is described by a limited set of commands sent to the robot. The first two datasets are uncompressed and no preprocessing is performed. **3) Kinetics600.** The last dataset is the Kinetics600 dataset (Kay et al., 2017) which is a diverse set of YouTube videos depicting human actions. The dataset is cropped and downsampled, which removes compression artifacts, to $64 \times 64$.

**Metrics.** Evaluation is based on bit rate in bits per pixel (bpp) and distortion measured in average frame peak signal-to-noise ratio (PSNR), which is related to the frame mean square error. In the supplementary material, we also report on multi-scale structural similarity (MS-SSIM) (Wang et al., 2004) which is a perception-based metric that approximates the change in structural information.

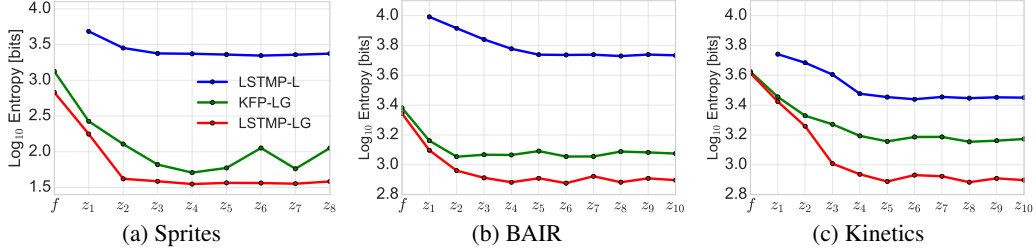

Figure 4: Average bits of information stored in $\boldsymbol{f}$ and $\boldsymbol{z}_{1:T}$ with PSNR 43.2, 37.1, 30.3 dB for different models in (a, b, c). Entropy drops with the frame index as the models adapt to the sequence.

**Comparisons.** We wish to study the performance of our proposed local-global architecture with LSTM prior (LSTMP-LG) by comparing to other approaches. To study the effectiveness of the global state, we introduce our baseline model LSTMP-L which has only local states with LSTM prior $p_{\boldsymbol{\theta}}(\boldsymbol{z}_t \mid \boldsymbol{z}_{<t})$. To study the efficiency of the predictive model, we show our baseline model KFP-LG which has both global and local states but with a weak predictive model $p_{\boldsymbol{\theta}}(\boldsymbol{z}_t \mid \boldsymbol{z}_{t-1})$, a deep Kalman filter (Krishnan et al., 2015). We also provide the performance of H.264, H.265, and VP9 codecs. Traditional codecs are not optimized for low-resolution videos. However, their performance is far superior to neural or classical image compression methods (applied to compress video frame by frame), so their performance is presented for comparison. Codec performance is evaluated using the open source FFMPEG implementation in constant rate mode and distortion is varied by adjusting the constant rate factor. Unless otherwise stated, performance is tested on videos with 4:4:4 chroma sampling and on test videos with $T = 10$ frames. Comparisons with classical codec performance on longer videos is shown in the supplementary material.

## 4.2 Quantitative Analysis: Rate-Distortion Tradeoff

Quantitative performance is evaluated in terms of rate-distortion curves. For a fixed quality setting, a video codec produces an average bit rate on a given dataset. By varying the quality setting, a curve is traced out in the rate-distortion plane. Our curves are generated by varying $\beta$ (Eq. 7).

The rate-distortion curves for our method, trained on three datasets and measured in PSNR, are shown in Fig. 3. Higher curves indicate better performance. From the Sprites and BAIR results, one sees that our method has the ability to dramatically outperform traditional codecs when focusing on specialized content. By training on videos with a fixed content, the model is able to learn an efficient representation for such content, and the learned priors capture the empirical data distribution well. The results from training on the more diverse Kinetics videos also outperform or are competitive with standard codecs and better demonstrate the performance of our method on general content videos. Similar results are obtained with respect to MS-SSIM (supplementary material).

The first observation is that the LSTM prior outperforms the deep Kalman filter prior in all cases. This is because the LSTM model has more context, allowing the predictive model to be more certain about the trajectory of the local latent variables, which in turn results in shorter code lengths. We also observe that the local-global architecture (LSTMP-LG) outperforms the local architecture (LSTMP-L) on all datasets. The VAE encoder has the option to store information in local or global variables. The local variables are modeled by a temporal prior and can be efficiently stored in binary if the sequence $\boldsymbol{z}_{1:T}$ can be sequentially predicted from the context. The global variables, on the other hand, provide an architectural approach to removing temporal redundancy since the entire segment is stored in one global state without temporal structure.

During training, the VAE learns to utilize the global and local information in the optimal way. The utilization of each variable can be visualized by plotting the average code length of each latent state, which is shown in Fig. 4. The VAE learns to significantly utilize the global variables even though $\dim(\boldsymbol{z})$ is sufficiently large to store the entire content of each individual frame. This provides further evidence that it is more efficient to incorporate global inference over several frames. The entropy in the local variables initially tends to decrease as a function of time, which supports the benefits from our predictive models. Note that our approach relies on sequential decoding, prohibiting a bi-directional LSTM prior model for the local state.

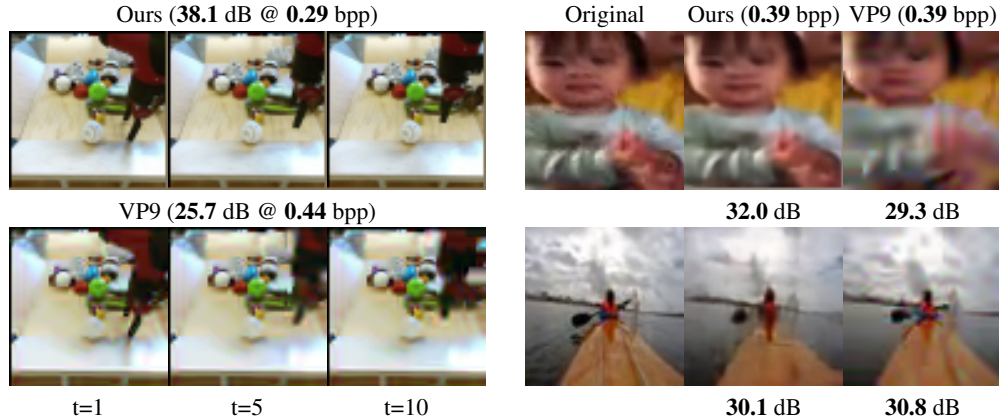

Figure 5: Compressed videos by our LSTMP-LG model and VP9 in the low bit rate regime (measured in bpp). Our approach achieves better quality (measured in dB) on specialized content (BAIR, left) and comparable visual quality on generic video content (Kinetics, right) compared to VP9.

## 4.3 Qualitative Results

We have shown that a deep neural approach (LSTMP-LG architecture) can achieve competitive results with traditional codecs with respect to PSNR or MS-SSIM (see supplementary material) metrics overall on low-resolution videos. Test videos from the Sprites and BAIR datasets after compression with our method are shown in Fig. 1 and Fig. 5 (left), respectively, and compared to modern codec performance. Our method achieves a superior image quality at a significantly lower bit rate than H.264/H.265 and VP9 on these specialized content datasets. This is perhaps expected since traditional codecs cannot learn efficient representations for specialized content. Furthermore, fine-grained motion is not accurately predicted with block motion estimation. The artifacts from our method are displayed in Fig. 5 (right). Our method tends to produce blurry video in the low bit-rate regime but does not suffer from the block artifacts present in the H.265/VP9 compressed video.

## 5 Conclusions

We have proposed a deep generative modeling approach to video compression. Our method simultaneously learns to transform the original video into a lower-dimensional representation as well as the temporally-conditioned probabilistic model for entropy coding. The best performing proposed architecture splits up the latent code into global and local variables and yields competitive results on low-resolution videos. For video sources with specialized content, deep generative video coding allows for a significant increase in coding performance, as our experiment on BAIR suggests. This could be interesting for transmitting specialized content such as teleconferencing.

Our experimental analysis focused on small-scale videos. One future avenue is to design alternative priors that better scale to full-resolution videos, where the dimension of the latent representation must scale with the resolution of the video in order to achieve high quality reconstruction. For the local/global architecture that we investigated experimentally, the GPU memory limits the maximum size of the latent dimension due to the presence of fully-connected layers to infer global and local states. While being efficient for small videos in the strongly compressed regime, this effectively limits the maximum achievable image quality. Future architectures may focus more on fully convolutional components. Besides a different temporal prior, the proposed coding scheme will remain the same.

Since our approach uses a learned prior for entropy coding, this suggests that improved compression performance can be achieved by improving video prediction. In future work, it will be interesting to see how our model will work with more efficient predictive models for full-resolution videos. It is also interesting to think about comparisons between deterministic and stochastic approaches to neural compression. We argue that by modeling the full data distribution of each frame, a probabilistic approach should be able to achieve shorter code lengths for fat-tailed and skewed data distributions than maximum-likelihood based compression methods. Thus we think that our work is a first step into a new direction for video coding which opens up several exciting avenues for future work.

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
