[Supplementary Material · Compressing_Low_Resolution_Videos_with_Sequential_Deep_Generative_Models.pdf]

# Supplemental Material:
# Deep Generative Video Compression

**Jun Han**[*]
Dartmouth College
junhan@cs.dartmouth.edu

**Salvator Lombardo**[*]
Disney Research LA
salvator.d.lombardo@disney.com

**Christopher Schroers**
DisneyResearch|Studios
christopher.schroers@disney.com

**Stephan Mandt**
University of California, Irvine
mandt@uci.edu

## 1 Model architecture

The specific implementation details of our model are now described. We describe the two baseline models, LSTMP-L and KFP-LG, and the best-performing LSTMP-LG model.

**LSTMP-L.** Our proposed baseline model LSTMP-L contains only local latent variables $z_t$ (the global state $f$ is omitted). This model is introduced to study the efficiency of the global state for removing temporal redundancy. The local state for each frame $z_t$ is inferred from each frame $x_t$. LSTMP-L employs similar encoder and decoder architectures as Ballé et al. (2016). The encoder $\mu_\phi(x_t)$ infers each $z_t$ independently by a five-layer convolutional network. For layer $\ell = 1$, the stride is 4, while a stride of 2 is used for layer $\ell = 2, 3, 4, 5$. The padding is 1 and the kernel size is $4 \times 4$ for all layers. The number of filters used for the Sprites video, for $\ell = 1, 2, 3, 4, 5$, are 192, 256, 512, 512 and 1024, respectively. For the more realistic video (BAIR and Kinetics video), the number of filters used at layer $\ell = 1, 2, 3, 4, 5$ are 192, 256, 512, 1024 and 2048, respectively. The decoder $\mu_\theta(z_t)$ is symmetrical to the encoder $\mu_\phi(x_t)$. With this architecture, the dimension of the latent state $z_t$ is 1024 for Sprites and 2048 for BAIR and Kinetics video. The prior for the latent state corresponding to the first frame, $p_\theta(z_1)$, is parametrized by the same density model defined on Appendix 6.1 of Ballé et al. (2018). The conditional prior $p_\theta(z_t \mid z_{<t})$ is parameterized by a normal distribution convolved with uniform noise. The means and (diagonal) covariance of the normal distribution are predicted by an LSTM with hidden state dimension equal to the dimension of the latent state $z_t$.

**LSTMP-LG.** LSTMP-LG is our proposed model in this paper which uses an efficient latent representation by splitting latent states into both global states and local states as well as the use of an effective LSTM predictive model for entropy coding. Now we describe the inference network. The two encoders $\mu_\phi(x_{1:T})$ and $\mu_\phi(x_t)$ begin with a convolutional architecture to extract feature information. The global state $f$ is inferred from all frames by processing the output of the convolutional layers over $x_{1:T}$ with a bi-directional LSTM architecture (note this LSTM is used for inference not entropy coding). This allows $f$ to depend on features from the entire segment. For the local state, the individual frame $x_t$ is passed through the convolutional layers of $\mu_\phi(x_t)$ and a two-layer MLP infers $z_t$ from the feature information of the individual frame. The decoder $\mu_\theta(z_t, f)$ first combines $(z_t, f)$ with a multilayer perceptron (MLP) and then upsamples with a deconvolutional network. The prior models $p_\theta(f)$ and $p_\theta(z_1)$ are parametrized by the density model defined in Appendix 6.1 of Ballé et al. (2018). The conditional prior $p_\theta(z_t \mid z_{<t})$ in the LSTMP-LG architecture is modeled by a normal distribution which is convolved with uniform noise. The means and covariance of the normal distribution are predicted by an additional LSTM.

---

[*] Shared first authorship.

(a) $p_{\boldsymbol{\theta}}(f^i)$  (b) $p_{\boldsymbol{\theta}}(f^i)$  (c) $p_{\boldsymbol{\theta}}(z_1^i)$  (d) $p_{\boldsymbol{\theta}}(z_1^i)$

Figure 1: Empirical distributions of the posterior of inference model and ground truth prior model in one specific rate-distortion BAIR example.

Both encoders $\boldsymbol{\mu}_{\boldsymbol{\phi}}(\cdot)$ have 5 convolutional (downsampling) layers. For layer $\ell = 1, 2, 3, 4$, the stride and padding are 2 and 1, respectively, and the convolutional kernel size is $4 \times 4$. The number of channels for layer $\ell = 1, 2, 3, 4$ are 192, 256, 512, 1024. Layer 5 has kernel size 4, stride 1, padding 0, and 3072 channels. The decoder architecture $\boldsymbol{\mu}_{\boldsymbol{\theta}}$ is chosen to be asymmetric to the encoder with convolutional layers replaced with deconvolutional (upsampling) layers. For the Sprites toy video, the dimensions of $\boldsymbol{z}$, $\boldsymbol{f}$, and hidden state $\boldsymbol{h}$ are 64, 512 and 1024, respectively. For less sparse videos (BAIR and Kinetics600), the dimensions of $\boldsymbol{z}$, $\boldsymbol{f}$, and LSTM hidden state $\boldsymbol{h}$ are 256, 2048 and 3072, respectively.

**KFP-LG.** KFP-LG is also a proposed baseline model which incorporates both the global state $\boldsymbol{f}$ and local latent $\boldsymbol{z}_t$ but uses a weaker deep Kalman filter predictive model $p_{\boldsymbol{\theta}}(\boldsymbol{z}_t \mid \boldsymbol{z}_{t-1})$ for entropy coding. The main purpose of the KFP-LG model is to compare to the LSTMP-LG model which has a longer memory. The conditional prior $p_{\boldsymbol{\theta}}(\boldsymbol{z}_t \mid \boldsymbol{z}_{t-1})$ in KFP-LG is described by a normal distribution with mean and variance that are parametrized by a three-layer MLP. The dimension at each layer of MLP is the same as the dimension of the latent state $\boldsymbol{z}_t$. KFP-LG has the same encoder and decoder structures as the proposed LSTMP-LG model aforementioned. The only difference between KFP-LG and LSTMP-LG is that they employ different prior models for conditional entropy coding.

## 2 Latent variable distribution visualization

In this section, we visualize the distribution of our prior model and compare to the empirical distribution of the posterior of the inference model estimated from data. In Fig. 1, we show the learned priors and the empirically observed posterior over two dimensions of the global latent state $\boldsymbol{f}$ and local latent state $\boldsymbol{z}$ in order to demonstrate that the prior is capturing the correct empirical distribution. From Fig. 1, we can see that the learned priors $p_{\boldsymbol{\theta}}(\boldsymbol{f})$ and $p_{\boldsymbol{\theta}}(\boldsymbol{z}_1)$ match the empirical data distributions well, which leads to low bit rate encoding of the latent variables. As the conditional probability model $p_{\boldsymbol{\theta}}(\boldsymbol{z}_t \mid \boldsymbol{z}_{<t})$ is high-dimensional, we do not display this distribution.

## 3 Additional performance evaluation

**MS-SSIM metric.** In the main paper, we evaluated performance in terms of PSNR distortion. Here, we also plot the MS-SSIM with respect to the bit rate to quantitatively compare our models to traditional codecs with respect to a perceptual metric. From Fig. 2, we can see that our LSTMP-LG saves significantly more bits when trained on specialized content videos and achieves competitive result when trained on general content videos.

**Longer videos.** We trained and evaluated our method on short video segments of $T = 10$ frames and evaluated classical codec performance on the same segments. However, for typical videos, somewhat longer segments tend to have less information per pixel than $T = 10$ segments, and standard video codecs are designed to take advantage of this fact. For this reason, we have presented video codec performances, evaluated on $T = 10, 30$, and 100 frame segments for the Kinetics data in Fig. 3. While existing codec performance improves for longer segments, we note that our method (trained and evaluated on 10 frames) is still comparable to modern codec performance evaluated on longer segments. Additionally, with proper design and training on longer video segments, our method could

be scaled to achieve similar temporal performance scaling since longer segments typically have less information per pixel.

Figure 2: Rate-distortion curves on three datasets measured in MS-SSIM (higher corresponds to lower distortion). Legend shared. Solid lines correspond to our models, with LSTMP-LG proposed.

Figure 3: Rate-distortion curves on the Kinetics dataset measured in PSNR. Codec performance is evaluated on video segments of $T = 10$, 30, and 100 frames. Our best performing method (trained and evaluated on $T = 10$ frames) is shown in red for comparison.