[Reviews · NeurIPS 2019]

Reviewer 1



The paper explores video compression learned end-to-end, which is relatively unexplored in the literature, and likely to be impactful for real world applications. The proposed method is an extension of the model proposed in [1] to the video setting. The author propose to add a global variable that captures the information about the sequence of frames in the video. Local (frame) and global (video) variables are trained by well-known techniques in amortized variational inference by recurring to parametric function approximators. The method is well executed and experiments show that the global variable helps in achieving, in the datasets presented, lower distortion for any given compression rate. The paper is well written and clear and it is overall enjoyable read. Pros: + The paper deals with a relatively unexplored domain. + The set of baselines is rather complete and experiment show the superiority of the proposed approach. Cons: - The methodological novelty is limited as the model is a straightforward extension of [1]. - The datasets are rather simplistic and the model is evaluated on videos long only 10 frames (which is quite short). Standard codecs seem to work better than the proposed model when applied to longer videos. In general, the experiments don't give a good idea of the applicability of the proposed model to more realistic settings (longer or higher resolution videos).

Reviewer 2



Originality: - Using deep learning methods for video compression is still underexplored and poses an interesting research direction compared to current (handcrafted) methods. - End-to-end framework: extension of a VAE with entropy coding to remove redundancy in the latent space -> combination of well-known DL model with previous work on image compression which is here applied on video data - (Uncited) recent work: 'DVC: An End-to-end Deep Video Compression Framework', Lu et al., CVPR19 Quality: - The use of a global encoding of the entire sequence might limit applicability of the approach, e.g., for encoding and transmitting live videos. Furthermore, the current approach seems to be limited to a small fixed sequence length. - The relation of Eq 3 wih Eq 4 is not obvious. Eq 3 only conditions on the z_t up to time t, while Eq 4 accesses all z_t of the sequence for the global encoding. - Evaluation demonstrates superior performance compared to traditional compression approaches on three datasets with varying degree of realism/difficulty. - Ablation study is provided which demonstrates the benefit of the model components (additional global representation for entire sequence, predictive model). Significance: - The combination of local and global feature is well motivated and the global feature is shown to have an significant impact on performance. However, the usability of the approach seems limited (small sequence length, global encoding of complete sequence). - The evaluation was performed only on short (10 frames) low resolution (64x64) videos. Superior results compared to traditional approaches were mainly achieved on special domain videos, the improvement on the diverse set Kinetics600 is relatively low and only evaluated within a small range of image quality scores. (Although the authors express their interest to examine the extension to full-resolution videos, it remains questionable whether this approach is feasible due to the high memory/GPU requirements.) Clarity: - Clear motivation for the approach (high internet traffic for videos required, usage of DL approach as promising alternative to current state-of-the-art) - l. 208, what is the time-dependent context c_t ? Minor: - If possible, figures should be shown on pages where they are mentioned. - Stated Fig. 5 in section 4.3 is missing or has been wrongly referenced. - first equation (p. 4) is not numbered - check references, e.g. - Higgins 2016, journal/ booktitel is missing - usage of abbreviations not consistent

Reviewer 3



Originality: While there does exist work in modeling video data with deep generative models, the authors are the first (to the best of my knowledge) to propose a neural, end-to-end video codec based on VAEs and entropy coding. The method offers a simple way to discretize the continuous latent space to learn a binary coding scheme of the compressed video. Although this has been explored in the context of image compression (e.g. Townsend 2019), it is important and useful. The generative model is actually quite similar in spirit to (Li & Mandt 2018), but with the added component of discretizing the latent space/entropy coding. Quality: The authors test their proposed method on 3 video datasets (Sprites, BAIR, Kinetics600), and evaluate their results using a variety of metrics (bpp, PSNR, MS-SSIM). Because there do not exist baselines to compare their method against, the authors provide a series of baselines to test the effect of each component of their model. The authors also clearly state the limitations of their method (GPU memory limitations with respect to the resolution at which they can compress videos, etc.). Although 64x64 sized videos are small, I believe this method is a great starting point for future work. Clarity: The paper was well-written, self-contained, and easy to follow, which I appreciated. The presentation of the model was clear as well. Significance: As video data comprises a significant proportion of the modern-day communications data in the Internet, the impact of this work is indeed significant. ---------------------------------------- UPDATE: Although I appreciated the authors' feedback, I wished they had addressed more of my questions in the Improvements section (e.g. regarding the strange plots with the bitrates and disentangled representations). However, as the authors noted that they will include additional experiments on longer video sequences in the final version, which is something I was particularly concerned with, I will keep my score as is.

[Author Response · NeurIPS 2019]

**General response (***R1, R2, R3***)**

Dear Reviewers, we thank you for taking the time to provide valuable feedback. We will correct the final manuscript to fix issues related to typos and missing citations. We strongly believe in the significance of our work as a first deep probabilistic modeling approach towards end-to-end video compression. Below we address the main issues raised.

**Simplicity of datasets and scalability.** First and foremost, we address comments (*R1, R2, R3*) regarding the simplicity of the datasets, being short, low-res videos. Our approach to performing video compression by entropy coding the video according to dynamic probabilities from a deep generative model (i.e. not block motion based video compression) is novel. Its performance depends on our ability to predict the distribution over future frames with low entropy. Currently all papers on deep video generation (upon which our approach is based) consider data sets comparable to ours and face scalability difficulties. Scalability limitations of the considered architecture can be can be partially ameliorated by using convolutional architectures instead of fully connected ones. We stress that our contribution is not only a specific sequential VAE architecture, but the full concept of using a temporally-conditioned, learned prior for entropy coding sequences, and gives rise to various extensions (multi-modal sequential priors, implicit distributions, hierarchical latent sequence models with multi-scale dynamics, etc.). We will emphasize these aspects more in a revised version.

**Relationship to existing image compression work.** While all reviewers agree that our approach extends existing work, we here clarify the similarities and differences to to Balle's approach to image compression, addressing in particular *R1*'s concern. While giving full credits to this related work for adopting its idea of discretization and entropy coding, we stress that our video compression model is quite different from Balle's image compression model, and that the sequential VAE setup that we analyzed posed new technical challenges that we overcame. In contrast to a stationary learned prior, our approach builds on sequential VAE architectures for high dimensional time series and in employs RNNs to model dynamics in the latent space. Further contributions include separating static from dynamic information with local/global variables as well as entropy-coding the sequence according to the non-stationary, learned prior. We compare with a basic temporal predictive model using Kalman filtering to show the advantages of such tailored design.

*R1* **Response:**

>...how the model could be scaled to longer videos and reported the results of this preliminary investigation...

Note that the video length is not an issue if one weakens the assumption of a global state besides the local one. We have recently found some efficient architectures that can be incorporated into our encoder/decoder part to improve our model. We are currently working on this and hope to include our results in the final revised version. Research progress towards better deep generative models for high-resolution, diverse video is required to push this idea further to high resolution. In particular, a generative model that outperforms next-frame block motion prediction on high-res video is needed.

*R2* **Response:**

>The use of a global encoding of the entire sequence might limit applicability of the approach...

Live videos can be divided into chunks of $T$ frames, where every chunk can be encoded separately with independent global states. The only limitation is that the video has to be encoded in chunks of $T$ frames, and during decoding a new reference global state must be used every $T$-frames. This is not too different from encoding/decoding chunks of video between key frames in classical codecs. The key frame stores information shared among the frames and is used as a reference to decode the subsequent frames. One advantage of the global encoding is that as a reference it has access to all of the frames in the chunk to better store shared information.

Note that the global state is not strictly required—our approach would also work with a purely local state. Our paper actually contains experiments with such architectures (see Fig. 3 LSTMP-L) as well, and is formulated more broadly.

>The relation of Eq 3 wih Eq 4 is not obvious...

The sequential prior is not specified in Eq. 4 as opposed to Eq. 3 (but is meant to be the same). We will fix this.

> (Uncited) recent work...

Thanks for pointing this out. We will cite and discuss it in our final version. We stress that this approach is different.

*R3* **Response:**

>... whether a deep probabilistic model for video compression is in fact better than a deterministic neural-network ...

At inference time, our method is deterministic as required by entropy coding. Our method is probabilistic in the sense that the probability distribution over all the next possible frames is used to remove temporal redundancy, and can be successively improved by better video prediction. Classical video codecs (or related neural network approaches) remove temporal redundancy by subtracting the most-likely estimate for the next frame from the actual next frame.

[Meta-Review · NeurIPS 2019]

This work builds upon image compression techniques based on variational auto-encoders (VAEs). The idea is novel, reasonably well explained and while the experiments are provided for low resolution, short video and low bitrates, the experiments do provide sufficient evidence that the proposed method is promising. As such the work is likely to be of interest to the community.